# Advances in Scale Assessment of Seabird Bycatch: A New Methodological Framework

**Dominik Marchowski** 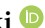

Ornithological Station, Museum and Institute of Zoology, Polish Academy of Sciences, ul. Nadwiślańska 108, 80-680 Gdańsk, Poland; dmarchowski@miiz.waw.pl

**Abstract:** This paper presents a methodology for indirectly estimating the scale of seabird bycatch using existing data. The study focuses on five key species of ducks that winter in the Polish waters of the Baltic Sea and are highly susceptible to bycatch: Long-tailed Duck, Velvet Scoter, Common Scoter, Greater Scaup, and Tufted Duck. The evaluation units used are divided into the Bornholm (BS) and Gotland Subdivisions (GS) within the Polish Exclusive Economic Zone (PEEZ). The analysis acknowledges the lack of bycatch data for certain areas known to have high concentrations of diving birds and fishing activity. The population sizes of waterbirds were assessed using ship-based surveys and a distance sampling approach. Fishing effort data from the five winter seasons between 2015/2016 and 2019/2020 were analyzed. Bycatch rates were estimated based on surveys conducted in previous seasons, and threshold values for bycatch were established using the concept of "small numbers" defined in EU directives. The results show that within the GS of the PEEZ, the estimated average abundance of all diving waterbirds was 174,800 individuals, with an average annual bycatch of 7921 birds (4.5% of the total). The Velvet Scoter was the most abundant species, followed by the Long-tailed Duck. In the BS, the estimated average abundance of diving waterbirds was 624,600 individuals, with an average annual bycatch of 5056 birds (0.8% of the total). The Long-tailed Duck was the most numerous species, followed by the Velvet Scoter. Acceptable bycatch thresholds were exceeded for all species in both subdivisions despite a much smaller scale of bycatch in the BS. The findings highlight the need for effective management and conservation measures to address the high mortality of seabirds due to bycatch. The methodology presented in this study offers a valuable approach for estimating bycatch scale and can support efforts to minimize the impact on seabird populations.

**Keywords:** waterbird bycatch; conservation implications; threshold values; bycatch estimation; Long-tailed Duck; Velvet Scoter; Greater Scaup; fishery

## 1. Introduction

Bycatch, which is a common occurrence in fishing activities, refers to the unintentional capture of non-target animals, such as marine mammals, sea turtles, non-target fish, and birds, aside from the desired catch [1–4]. This incidental capture of non-target species poses a significant threat to seabird populations, contributing to their mortality [5,6]. Waterbirds are susceptible to entanglement in various types of fishing gear and drowning. These species have long lifespans and low reproductive rates. As a result, their populations are vulnerable to losses, particularly among adult individuals, as it takes a relatively long time to recover from such losses [7].

Fishermen employ various fishing methods, with trawls, longlines, and gillnets being the primary techniques used [8]. Among these, gillnets are particularly concerning as they can inadvertently trap and kill protected, rare, or threatened species [9]. While gillnet fishing has been prohibited in open ocean waters [8], it persists in shallow shelf seas, bays, and lagoons, leading to the depletion of protected seabird populations. It is estimated that at least 400,000 birds lose their lives annually due to gillnets [9].

The southern Baltic Sea is among the three regions worldwide with the highest gillnet bycatch rates, alongside the north-west Pacific and Iceland [9]. This situation arises from the significant overlap between the spatial and temporal distribution of diving birds and gillnet fisheries [6,10]. The issue of gillnet bycatch in wintering seabird hotspots within the Baltic Sea and adjacent areas of the North Sea was identified several decades ago. However, the available data on this topic were often limited in terms of spatial and temporal coverage [11–15]. Bycatch predominantly occurs during the winter and seabird migration seasons in the Baltic Sea [9]. Many seabird species, including sea ducks, divers, and auks, which primarily nest in the Arctic, spend their winters in the Baltic Sea [16,17].

Recent analyses indicate that the Polish waters of the Baltic Sea were one of the most significant hotspots where a large number of seabirds have perished [4]. This can be attributed to the convergence of two factors: favorable fishing conditions and a relatively large gillnet fishing fleet [4], as well as excellent wintering conditions for a substantial population of seabirds, thanks to shallow areas rich in food resources [18,19]. It has been estimated that the highest seabird bycatch in the Polish Exclusive Economic Zone occurred in the 1970s, with around 47,000 birds perishing annually. However, the scale of bycatch has gradually decreased, reaching approximately 40,000 birds per year in the 1980s and 1990s, and about 20,000 birds per year in the 2010s [4].

The reduction in bycatch scale over the past few decades can be attributed to several factors. Primarily, the wintering bird population has significantly declined, experiencing a 50% decrease over the last 30 years [19]. Additionally, the size of the fishing fleet has also diminished [20]. The most commonly caught bird species within the Polish Exclusive Economic Zone include the Long-tailed Duck, Velvet Scoter, Greater Scaup and Tufted Duck with the first two being classified as vulnerable (VU category) on the IUCN Red List [21,22]. Moreover, all four species are listed on the Red List of the Helsinki Commission for the Baltic Sea [23].

There are several ways to monitor bycatch. The basic method involves reporting incidents by fishermen [24]. However, studies have shown that fishermen tend to underestimate and hesitate to report bycatch reliably [15]. Anonymous surveys of fishermen [25] or analysis of birds caught and brought to the port [4,26] are alternative approaches for monitoring bycatch. However, both methods rely on fishermen's cooperation, which can introduce bias into the results [15].

Bycatch is considered undesirable by fishermen, who commonly discard drowned birds overboard [27]. These birds often wash up on beaches over time and counting them provides another means of assessing the extent of waterbird bycatch [28]. This method has limitations, as it is uncertain whether all caught birds were treated this way by fishermen or if they were all washed ashore [29].

A more reliable approach to monitor bycatch is to have independent observers on fishing boats who record bycatch incidents. This method enables a more accurate assessment of bycatch scale and allows for calculation of bycatch rates with an appropriate sample size [30]. Electronic monitoring is also an effective and increasingly popular method [31]. However, it may not be feasible to install electronic systems on small fishing vessels [25].

To ensure accurate bycatch rates, research relying on independent observers or electronic monitoring is recommended. However, monitoring bycatch is a costly undertaking, and regular implementation is uncommon in most countries. Even if official government reports exist, they often significantly underestimate the true extent of the issue [32]. Consequently, there is a need to develop a methodology that enables the assessment of bycatch scale indirectly. While there are records of fishing effort provided by fishermen [24], they seldom report instances of bycatch in these records [25]. Additionally, data on seabird abundance and distribution were available because of monitoring conducted for reporting purposes under European Union directives such as the Birds Directive or the Marine Strategy Framework Directive [33]. Local surveys have also been conducted to calculate bycatch rates, including surveys in the southern Baltic area conducted in Germany [15], Lithuania [32,34,35], and Poland [25].

The objective of this paper is to present a methodology for indirectly estimating the scale of seabird bycatch using the existing data, thus allowing for seabird bycatch assessment in regions with poor monitoring schemes. The methodological approach was proposed for five species of sea ducks that winter in significant numbers in the Polish waters of the Baltic Sea and are also highly susceptible to bycatch in fishing nets: Long-tailed Duck (*Clangula hyemalis*), Velvet Scoter (*Melanitta fusca*), Common Scoter (*Melanitta nigra*), Greater Scaup (*Aythya marila*), and Tufted Duck (*Aythya fuligula*).

## 2. Methods

In the Polish Exclusive Economic Zone (PEEZ), which includes coastal waters and coastal lagoons spanning approximately 30,500 km², five species of benthivorous ducks (Long-tailed Duck, Velvet Scoter, Common Scoter, Greater Scaup and Tufted Duck) can be found. These species are all listed as threatened according to HELCOM Red List [23]. The evaluation units used for waterbird indicators in this area are divided into the Bornholm Subdivision and the Gotland Subdivision [36] (Figure 1). It should be noted that the analysis lacks data for a significant portion of Lake Dąbie and the lower sections of the Odra and Vistula rivers, which are known to have a high concentration of diving birds [37]. These areas are also exploited for fishing, making them likely hotspots for bird bycatch [4].

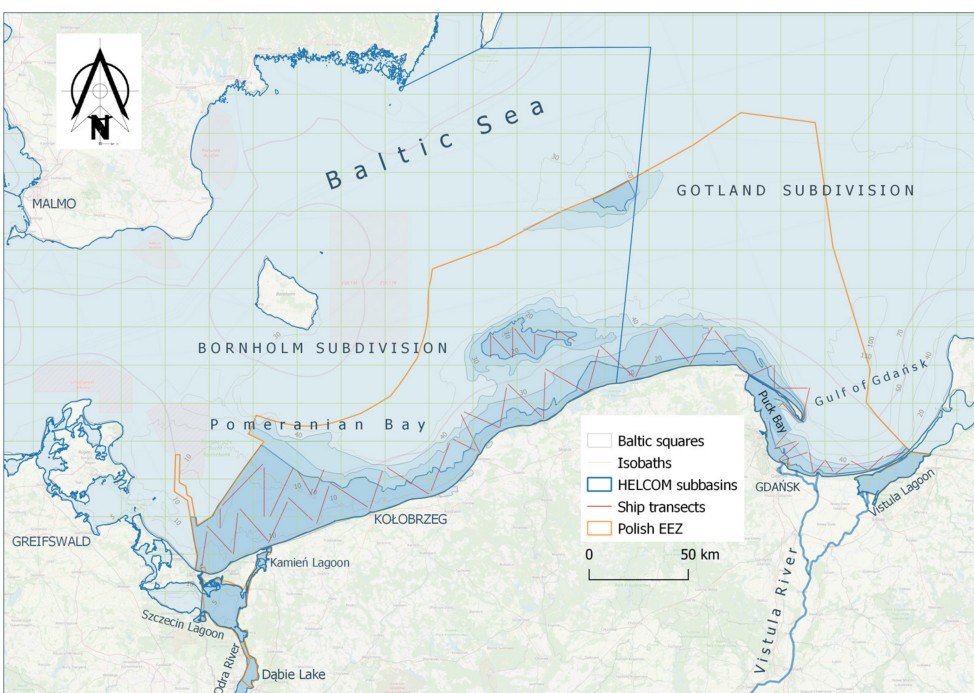

**Figure 1.** The study area—Polish Exclusive Economic Zone in the southern part of the Baltic Sea, divided into the HELCOM subdivisions (Bornholm and Gotland) and Baltic Squares (20 × 20 km). Red lines are the transects used for ship-based monitoring of waterbirds offshore.

### 2.1. Waterbird Abundance Assessment

The waterbird data were analyzed for the five winter seasons spanning from 2015/2016 to 2019/2020. The number of waterbirds was obtained from annual January counts and assumed to remain constant throughout the winter period (from 1 October to 30 April), this included Long-tailed Duck, Velvet Scoter, and Common Scoter. In the case of Greater Scaup and Tufted Duck, which reach higher numbers in autumn and spring [10,16], the numbers were averaged from three counts in the winter season of November, January, and March. Waterbird populations were assessed through ship-based surveys conducted along 56 transects, ranging from 3.9 km to 28.7 km in length [38] (Figure 1). These surveys

followed a standardized study protocol [39,40] and utilized a distance sampling approach, a widely accepted method in seabird research [41,42].

The same linear transects were utilized consistently over five consecutive years of research. The surveys were conducted under favorable wind conditions, ranging from 0 to 3 on the Beaufort scale, with the vessel maintaining a constant speed of approximately 8 knots. Dual observers simultaneously conducted the observations, with one responsible for counting birds on the left side and the other on the right [43]. The observers changed regularly, with a minimum of three observers present during the survey. While two observers worked, one or more rested. Fieldwork was conducted using two vessels, and the observation bridge was positioned at heights of 2.7 and 3.4 m above the water surface. To determine the distance of birds from the ship, the transparent ruler method was employed. Horizontal lines were marked on the ruler, and the birds were counted within the following distance ranges: A—0–50 m, B—50–100 m, C—100–200 m, and D—200–300 m. Observations beyond the transect, at distances exceeding 300 m, were categorized as sector E [43]. The distances to birds sitting on the water were individually adjusted for each observer and vessel, employing the Heinemann equation [44].

$$D = \left( \frac{A \times B(3838 \times B^{0.5} - C)}{B^2 + 3838 \times B^{0.5} \times C} \right) \times 100$$

where, A represents the length of the observer's arm from the eye to the ruler in meters, B denotes the height of the observer's eye from the water surface (deck height + distance from the ground to the observer's eye), and C refers to the transect sector distance.

In the analysis, the Conventional Distance Sampling (CDS) method was used with the inclusion of a log transformation for the 'size' (cluster size) variable in the model. This transformation was chosen to address any departure from normality and improve the modeling assumptions [45].

Multiple Covariate Distance Sampling (MCDS) was utilized, and additional covariates such as 'Obs' (observer) and 'sea.state' (sea state) were incorporated into the model to account for their potential effects on the detection probability and abundance estimation. The analysis considered right truncation to accommodate any potential bias resulting from individuals observed solely within a specific distance range. The delta-method approximation was employed to calculate the bounds of the confidence intervals, assuming a log-normal sampling distribution of the estimated abundance [46].

To determine the best-fitting function, key parametric functions were evaluated, including half-normal, and hazard rate. The selection of the most suitable function was based on the lowest Akaike Information Criterion (AIC) values [47]. Data analyses were conducted in the R environment (R Core Team 2021) using the Distance package (version 1.0.4) for Distance Sampling Detection Function and Abundance Estimation [45]. The general mathematical formula for the Distance Sampling (DS) analysis is as follows:

$$N\_ds = (M\_ds \times A\_ds)/P\_ds$$

The estimated population abundance using the Distance Sampling method is calculated by multiplying the mean group size (M_ds) by the effective surveyed area (A_ds) and dividing it by the probability of detection (P_ds). This method accounts for the detectability of individuals within the surveyed area.

Additionally, the bird counts obtained from shore-based surveys conducted during the standard January waterbird count and the International Waterbird Census (IWC) [40] were incorporated into the estimated offshore counts. The offshore data collected using the Distance method, as described above, corresponds to birds observed at a distance of >1 km from the shore. Conversely, the IWC data includes birds present both at the shoreline and up to 1 km from the shore, as well as birds observed in internal sea waters such as lagoons and shallow bays. In the analysis, the results from both methods were aggregated by combining them because they pertain to different subsets of the bird population but

originate from the same overall population. The Distance estimation specifically focuses on offshore birds, while the IWC data encompasses coastal and internal marine water birds (refer to the formulas below).

International Waterbird Census (IWC):

$$N\_iwc = \Sigma(N\_i)$$

The estimated population abundance in coastal waters, obtained using the IWC method [40], is calculated by summing up the observed population abundances for each specific section or object within the coastal area. This method involves one observer counting all the birds within their assigned section, which typically spans approximately 10–15 km of coastline, or counting birds within an entire facility such as a lagoon, or sea bay. The results obtained from each section and object are then aggregated.

Total Population Abundance (N_sum):

$$N\_sum = N\_ds + N\_iwc$$

The total population abundance is obtained by summing up the population abundances estimated using the Distance Sampling method (N_ds) and the population abundances obtained from coastal waters using the IWC method (N_iwc). This provides a comprehensive assessment of the overall population abundance by combining data from both methods. The results of estimating the number of birds in individual seasons are included in the Supplementary Materials (Table S1).

### 2.2. Fishing Effort

The fishing effort data were analyzed for the five winter seasons spanning from 2015/2016 to 2019/2020. In order to assess the overall fishing activity, an examination of data obtained from the Fisheries Monitoring Centre (CMR) was conducted. Specifically, data from the winter period (October–April) were considered, as this is when the highest occurrence of seabirds is observed in this particular area of the Baltic Sea [19].

The data on fishing catches included various variables, such as the fishing vessel ID, length of the fishing vessel, names of the port of departure and return, type of fishing gear, target species of fish, number of fishing gear deployed, fishing duration, and fishing location (Baltic square, Figure 1).

The analysis focused solely on static nets, which are known to be the most problematic fishing gear in terms of bird bycatch in this region [4,9,14]. Other fishing gears, such as longlines and fyke traps, which may incidentally capture waterbirds but are considered less problematic than static nets in terms of bird bycatch within the study area, were not taken into account due to limited available fishing effort and/or bycatch rate data for these gears [4]. It is important to note that the analysis did not consider the impact of Polish vessels operating outside of Polish waters or within the Exclusive Economic Zones of other countries, nor did it account for non-Polish fleets operating within the Polish Exclusive Economic Zone.

For each fishing record, the standard unit of fishing effort was calculated in net*meter*days (NMD) [15,25]. The NMD unit represents the combined length of nets deployed in the water over a specific number of days, reflecting the duration during which they posed a potential threat to birds.

### 2.3. Seabird Bycatch

Using the bycatch rates that indicate the number of birds caught per 1000 NMD in the primary static net fisheries operating in Polish sea waters (Table 1), the overall bycatch count for all species was estimated by multiplying these rates with the total fishing effort for each year at the Statistical Baltic Squares (SKB) level. Each SKB covers an offshore area of approximately 400 km$^2$, while the squares adjacent to the coast, extending beyond the borders of the PEEZ, are smaller in size (Figure 1). Subsequently, the total mortality due

to bycatch for individual species was determined by calculating the proportion of their specific bycatch mortality in relation to the overall waterbird population. The bycatch rates were computed based on surveys conducted in the winter seasons of 2013/2014 and 2014/2015, involving observers onboard fishing vessels in water bodies within the PEEZ, including Kamień Lagoon, Szczecin Lagoon, Pomeranian Bay, and Puck Bay. These study sites were selected to represent the entire Polish fishery, encompassing areas with high, medium, and no observed bycatch [25].

**Table 1.** Bycatch rates based on studies carried out in the Polish waters of the Baltic Sea in the winter seasons of 2013/2014 and 2014/2015 (according to Psuty et al., 2017, [25]). Bycaught birds/1000 NMD: number of bycaught birds per 1000 nets × meters × days.

| Type of Static Nets | Bycaught Birds/1000 NMD (95% CI) |
|---|---|
| Cod, flounder, and turbot gillnets/trammel nets | 0.221 (0.218–0.225) |
| Herring, perch, roach, garfish and spart gillnets | 0.227 (0.217–0.238) |
| Zander and bream gillnets | 0.651 (0.447–1.386) |
| Trout, salmon, pike and whitefish gillnets and one-side anchored nets (i.e., semi-driftnets) | 0.279 (0.250–0.309) |

### 2.4. Setting the Threshold Values

To establish a benchmark for "zero bycatch", BirdLife International [48] proposed a threshold of 1% of the natural annual adult mortality. This acknowledges that even with the implementation of effective mitigation measures, it is likely that a small number of seabirds will still be incidentally caught. The 1% threshold is based on legal interpretations by the European Court of Justice regarding the concept of "small numbers" as defined in the EU Birds Directive and EU guide to sustainable hunting [49]. As determining the exact natural annual adult mortality for most species in the presence of anthropogenic causes is challenging, it is more practical to use the total annual adult mortality as an approximation. The annual adult mortality (*m*) is calculated from the survival rates (*s*) using the equation:

$$m = 1 - s \qquad (1)$$

Species-specific survival values for adult individuals, which are necessary for calculating mortality across all bird species, can be found in the literature, such as Bird et al. [50]. The species-specific threshold value (SST) is then estimated by multiplying the estimated population size (*N*) in the evaluation area by the species-specific annual adult mortality rate (m) and 1%:

$$\text{SST} = \hat{N} \times \text{m} \times 0.01 \qquad (2)$$

where, $\hat{N}$ (N_sum) represents the estimated population size, and *m* denotes the annual mortality rate of adults for the species or population.

### 3. Results

In total, within the Gotland subdivision of the Polish Exclusive Economic Zone (PEEZ), the estimated abundance of all diving waterbirds averaged 174,800 individuals yearly over the seasons from 2015/16 to 2019/20. During these seasons, the average bycatch estimates reached 7921 birds annually, accounting for 4.5% of the total. Among the species, the Velvet Scoter (*Melanitta fusca*) was the most abundant, with an average of 72,834 (Table S1) individuals (average annual bycatch = 3036). The Long-tailed Duck (*Clangula hyemalis*) ranked second in abundance, with an average of 49,606 individuals (Table S1)observed throughout the study period (average annual bycatch = 2135; Table 2). Benthivorous ducks dominated all ecological groups and constituted 86.0% of the total diving bird population in the area.

**Table 2.** The most abundant diving bird species present in the bycatch in the Polish part of Gotland subdivision, their abundance, and the scale of the bycatch (mean and confidence intervals). Mean for the seasons 2015/16–2019/20.

| Species | Abundance | Bycatch Mean | Bycatch 95%CI− | Bycatch 95%CI+ |
|---|---|---|---|---|
| Velvet Scoter | 72,834 | 3036 | 2444 | 4989 |
| Long-tailed Duck | 49,606 | 2135 | 1726 | 3475 |
| Tufted Duck | 14,414 | 594 | 479 | 985 |
| Common Scoter | 7804 | 353 | 281 | 593 |
| Greater Scaup | 5674 | 238 | 193 | 386 |

For the Bornholm subdivision within the Polish Exclusive Economic Zone (PEEZ), the estimated abundance of all diving waterbirds averaged 624,600 individuals yearly over the seasons from 2015/16 to 2019/20. The average bycatch for these seasons amounted to 5056 birds annually, which corresponds to 0.8% of the total population. Among the species, the Long-tailed Duck was the most numerous, with an average of 331,757 individuals (Table S1) (average annual bycatch = 2305). The Velvet Scoter ranked second in abundance, with an average of 167,324 individuals (Table S1) observed throughout the study period (average annual bycatch = 1318; Table 3). Benthivorous ducks dominated all ecological groups of diving birds and constituted 96.1% of the total bird population in the area.

**Table 3.** The most abundant diving bird species present in the bycatch in the Polish part of Bornholm subdivision, their abundance, and the scale of the bycatch (mean and confidence intervals). Mean for the seasons 2015/16–2019/20.

| Species | Abundance | Bycatch Mean | Bycatch 95%CI− | Bycatch 95%CI+ |
|---|---|---|---|---|
| Long-tailed Duck | 331,757 | 2305 | 2294 | 3113 |
| Velvet Scoter | 167,324 | 1318 | 1117 | 1469 |
| Common Scoter | 62,064 | 450 | 384 | 552 |
| Greater Scaup | 22,724 | 190 | 170 | 221 |
| Tufted Duck | 16,333 | 141 | 110 | 133 |

For two subdivisions the allowable bycatch thresholds were exceeded (Gotland subdivision - refer to Table 4, Bornholm subdivision refer to Table 5).

**Table 4.** Gotland, 1% threshold of mean wintering population abundance in Polish part of Gotland subdivision with an indication whether the threshold values have been exceeded or not. Mortality rate was taken from Bird et al., 2020 [50].

| Species | Mortality Rate | SST/Bycatch | Achievement (A) or Failure (F) of Threshold Values | Helcom Red List Status |
|---|---|---|---|---|
| Velvet Scoter | 0.21 | 153/3036 | F | Endangered |
| Long-tailed Duck | 0.25 | 124/2135 | F | Endangered |
| Tufted Duck | 0.29 | 42/594 | F | Near Threatened |
| Common Scoter | 0.22 | 17/353 | F | Endangered |
| Greater Scaup | 0.26 | 15/238 | F | Vulnerable |

**Table 5.** Bornholm, 1% threshold of mean wintering population abundance in Polish part of Bornholm subdivision with an indication whether the threshold values have been exceeded or not. Mortality rate was taken from Bird et al., 2020 [50].

| Species | Mortality Rate | SST/Bycatch | Achievement (A) or Failure (F) of Threshold Values | Helcom Red List Status |
|---|---|---|---|---|
| Long-tailed Duck | 0.25 | 829/2305 | F | Endangered |
| Velvet Scoter | 0.21 | 351/1318 | F | Endangered |
| Common Scoter | 0.22 | 137/450 | F | Endangered |
| Greater Scaup | 0.26 | 59/190 | F | Vulnerable |
| Tufted Duck | 0.29 | 47/141 | F | Near Threatened |

## 4. Discussion

As indicated by the Results, none of the assessed species have achieved a favorable status. The threshold value set by HELCOM as the core indicator in the holistic assessment of the ecosystem health of the Baltic Sea [36], following the recommendations of BirdLife International and HELCOM/OSPAR experts, which is based on a 1% adult mortality threshold (*SST*), has been exceeded for each species and within each HELCOM subdivision.

It is important to note that the assessment was conducted using average population sizes (2016–2020) observed exclusively in the Polish part of each subdivision, along with the corresponding estimated bycatch. The Bornholm subdivision exhibited higher bird abundances (Tables 2 and 3), while the bycatch rates were relatively lower compared to the Gotland subdivision (4.5% vs. 0.8% bycatch relative to abundance). This suggests that the Polish part of the Gotland subdivision experiences a higher density of fishing nets and greater fishing intensity compared to the Bornholm subdivision. The most significant hotspots where large concentrations of birds overlap with intense fishing activities in the Polish part of the Gotland subdivision are the Gulf of Gdańsk and Vistula Lagoon. In the Polish part of the Bornholm subdivision, the Odra Estuary (which includes Odra River, Dąbie Lake, Szczecin Lagoon and Kamień Lagoon), southern Pomeranian Bay, and the vicinity of Kołobrzeg are the key areas of concern [4].

It is worth noting that fishing effort in the Polish EEZ has decreased in recent years due to EU regulations prohibiting cod fishing [4]. These regulations are expected to reduce bird bycatch, as cod nets were responsible for a significant portion of bird bycatch in Polish sea waters [24].

The findings of this article clearly indicate that bycatch poses a significant threat to seabird populations. While there is an obligation to monitor bycatch, not all countries regularly conduct such studies [4], and the available official reports often significantly underestimate the magnitude of the problem [23]. Therefore, there is a need to develop a methodology that enables the assessment of bycatch scale indirectly, using the available data.

Within this analysis, various aspects were considered, including the fishing effort declared by fishermen, data on the abundance and distribution of seabirds, and bycatch rates calculated based on local surveys [15,24]. However, several important factors should be taken into account in the context of the discussed results.

First, it is crucial to have systematic monitoring of bycatch by governments and fisheries management institutions [28]. Only through regular studies will we have a comprehensive understanding of the problem's scale and be able to take appropriate actions for the protection of seabirds.

Second, it is important for fishermen to report cases of bycatch. Currently, it is often the case that fishermen do not include incidental catches in their declarations and discard them [51], leading to an underestimation of the magnitude of the problem [25]. Therefore, educational and awareness-raising efforts are necessary to increase fishermen's

understanding of the impacts of incidental bird catches and to encourage them to report such incidents [52].

In the Southern Baltic region, gillnets are the main cause of bycatch [4,14]. The method described in the article was specifically developed to address the impact of gillnet fishing on seabird populations. It takes into consideration the behavior of birds in the study area, such as their seasonal abundance patterns, which are characterized by high numbers in winter and minimal presence in summer [16]. Therefore, it is crucial to appropriately scale and adapt this method to local conditions in order to obtain accurate and realistic results. After that, this method can be successfully applied to other fishing fleets, such as those using longlines, provided that it is properly adapted to their specific conditions.

The methodological framework described in this article should be regarded as a means of continuous monitoring of bycatch. Typically, bycatch assessment is conducted on an occasional, ad hoc basis, both at local scales [25,26,29], larger scales [53], and globally [54]. This method should be complemented by bycatch monitoring using trained observers or electronic monitoring systems. Such monitoring is necessary to scale up this method, as updated bycatch rates are generated periodically, enabling annual assessments. By employing this approach, we can move away from sporadic surveys and obtain an annual overview of the phenomenon, allowing us to identify long-term trends.

Another significant point to consider is the need to increase efforts in seabird conservation and implement effective measures to reduce bycatch [55]. The adoption of innovative technological solutions and bird-friendly fishing practices can contribute to the reduction in incidental captures and minimize the impact of fishing activities on seabird populations [56].

Moreover, additional research is imperative to enhance our understanding of the factors influencing bycatch and to formulate effective management strategies [57]. Studies focusing on the effectiveness of bycatch mitigation measures, the identification of high-risk areas, and the assessment of the long-term impact of bycatch on seabird populations would be valuable contributions to the field [6,15].

To summarize, tackling the problem of seabird bycatch necessitates a collective endeavor involving governmental entities, fisheries management institutions, fishermen, and researchers [58]. Through the establishment of consistent monitoring practices, encouragement of reporting mechanisms, and implementation of impactful conservation measures, we can strive to minimize the repercussions of bycatch and safeguard the enduring existence of seabird populations in our marine environments.

## 5. Conclusions

This article should not be interpreted as an endorsement for governments of countries obligated to monitor bycatch to disregard such monitoring. On the contrary, regular monitoring should be carried out as it provides direct data on bycatch. However, the objective of this article was to encourage the reporting of bycatch scale using indirect data in situations where direct monitoring is unavailable. The intention was not to exploit the absence of bycatch monitoring as a means to avoid reporting bird mortality in fishing nets and, in turn, neglect the assessment of fishing's impact on the conservation status of marine avian species.

**Supplementary Materials:** The following supporting information can be downloaded at: https://www.mdpi.com/article/10.3390/d15070808/s1, Table S1: Estimated bird numbers obtained on the basis of Distance Sampling (DS) in Offshore areas and on the basis of counts carried out using the International Waterbird Census (IWC) method, in the area of the Polish Exclusive Economic Zone in the Baltic Sea, broken down into the Bornholm and Gotland Subdivisions.

**Funding:** The research was carried out as part of the statutory activities of the Museum and Institute of Zoology of the Polish Academy of Sciences, Warsaw, Poland.

**Institutional Review Board Statement:** Not applicable.

**Data Availability Statement:** Data is available from the author upon request.

**Acknowledgments:** The data used to create this publication were collected through government monitoring programs, specifically the Fisheries Monitoring Center for fishing effort data, and the Chief Inspectorate of Environmental Protection for Monitoring Wintering Waterbirds and Wintering Waterbirds of Transitional Waters. I would also like to thank Julius Morkūnas for sharing a photo of dead birds drowned in fishing nets in the southern Baltic Sea, the photo was used for the Graphic Abstract.

**Conflicts of Interest:** The author declares that he has no known competing financial interest or personal relationships that could have appeared to influence the work reported in this paper.

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
