# Peer review of "Advances in Scale Assessment of Seabird Bycatch: A New Methodological Framework"

_diversity, doi:10.3390/d15070808_

Round 1

Reviewer 1 Report

Revisão Artigo Diversity

Advances in scale assessment of seabird bycatch: a new methodological framework

Overall comments:

I consider this study of great potential to become a very-useful research piece. The article deals with a cutting-edge issue for seabirds’ conservation and contains very comprehensive information of great value. I believe it has noticeable relevance and interest for this journal's audience. However, I have concern about some aspects of the way the information is used. In my opinion, these issues should be addressed in order to raise the reader's confidence in the analysis and to highlight the novelty of these concluding results in the context of current knowledge about the impacts of fisheries on seabirds.

Line 9. Please rephrase or eliminate the statement: “eliminating the need for regular monitoring”. See next comment.

line 84. The author state in the goals of the paper that the present methodology “eliminating the need for regular monitoring”. This sentence is better explained in the discussion section (Lines 268-270), but I recommend rephrasing. As a suggestion you can say the methodology allows for seabird bycatch assessment in regions with poor monitoring schemes. Also, author should present in the introduction a brief state of the art regarding methods to access seabird bycatch, in order to, in the discussion be compared with the presented methodology.  

line 97,98. Lake Dąbie and Odra and Vistula rivers are referred in the text. Please add this location to Fig 1. If the reader is not familiarized with the study area, will not know where this lake and rivers are.

Line 101. Better to move the survey dates to section 2.1

Line 111. It’s not clear if the same survey lines were surveyed in each survey year. Please, add information about survey effort conditions, namely Beaufort or sea state and survey platform conditions (speed, observer rotation? observation bridge altitude). The survey conditions are referred to Chodkiewicz et al. 2012 (reference 32). If I correctly understood Chodkiewicz et al. 2012, reports to monitoring of Polish birds in 2010-2012. The present work deal with data from 2015/2016 to 2019/2022. Therefore, is better to add the requested information to clarify the data acquisition procedures.

Line 112. Did you use Conventional distance sampling (CDS), MCDS? Right truncation? Bootstrap? Please add more detailed information regarding distance analysis.

Line 121. it is said that “… the bird counts obtained from shore-based surveys…were incorporated into the estimated offshore counts”. Please give detailed information how did you process this data. Was incorporated into distance analysis? This need to be very well justified.

Line 123 and following lines. It’s said: “an examination of data obtained from 123 the Fisheries Monitoring Centre (CMR) was conducted”, what data? Number of vessels, deployed of fishing net coordinates, etc.? please add further details.

Line 177 and following lines. Please add Confidence intervals (CI) and Coefficient of variation (CV) to the estimated abundance values. This can be presented as a table in supplementary materials.

Discussion: should be added a paragraph discussing what is the novel of this methodological framework comparing with others bycatch approaches, e.g.  

https://doi.org/10.1111/1365-2664.13407; Report of the Final Global Seabird Bycatch Assessment Workshop - WCPFC-SC15-2019/EB-WP-07 (birdlife South Africa 2019); https://doi.org/10.1093/icesjms/fsw088; https://doi.org/10.1016/j.biocon.2022.109463

Author Response

Reviewer 1.

Advances in scale assessment of seabird bycatch: a new methodological framework

Overall comments:

I consider this study of great potential to become a very-useful research piece. The article deals with a cutting-edge issue for seabirds’ conservation and contains very comprehensive information of great value. I believe it has noticeable relevance and interest for this journal's audience.

However, I have concern about some aspects of the way the information is used. In my opinion, these issues should be addressed in order to raise the reader's confidence in the analysis and to highlight the novelty of these concluding results in the context of current knowledge about the impacts of fisheries on seabirds.

Reply: I would like to thank Reviewer 1 for positive feedback on my article, especially for the critical comments. I concentrated on them and tried to improve my text point by point according to the comments and suggestions.

Line 9. Please rephrase or eliminate the statement: “eliminating the need for regular monitoring”. See next comment.

Reply: Indeed, this statement may not be entirely correct. I admit I've wondered about that too. Probably best to just eliminate it, so I did in the new version of the text.

line 84. The author state in the goals of the paper that the present methodology “eliminating the need for regular monitoring”. This sentence is better explained in the discussion section (Lines 268-270), but I recommend rephrasing. As a suggestion you can say the methodology allows for seabird bycatch assessment in regions with poor monitoring schemes.

Also, author should present in the introduction a brief state of the art regarding methods to access seabird bycatch, in order to, in the discussion be compared with the presented methodology.

R: As suggested by the reviewer, the statement "eliminating the need for regular monitoring" was replaced with another: "allowing for seabird bycatch assessment in regions with poor monitoring scheme-schemes".

In the introduction, also as suggested by the reviewer, I presented a brief overview of the methods used for bycatch monitoring.

line 97,98. Lake Dąbie and Odra and Vistula rivers are referred in the text. Please add this location to Fig 1. If the reader is not familiarized with the study area, will not know where this lake and rivers are.

R: That's indeed right, I've corrected the map to make it more readable, added missing names.

Line 101. Better to move the survey dates to section 2.1

R: The text from lines 101-104 has been separated and moved to the appropriate places in section 2.1. and 2.2.

Line 111. It’s not clear if the same survey lines were surveyed in each survey year. Please, add information about survey effort conditions, namely Beaufort or sea state and survey platform conditions (speed, observer rotation? observation bridge altitude). The survey conditions are referred to Chodkiewicz et al. 2012 (reference 32). If I correctly understood Chodkiewicz et al. 2012, reports to monitoring of Polish birds in 2010-2012. The present work deal with data from 2015/2016 to 2019/2022. Therefore, is better to add the requested information to clarify the data acquisition procedures.

R: As suggested by the reviewer, I added more details to the methodology of data collection using Distance Sampling.

Line 112. Did you use Conventional distance sampling (CDS), MCDS? Right truncation? Bootstrap? Please add more detailed information regarding distance analysis.

R: In the analysis, I used the Conventional Distance Sampling (CDS) method with the inclusion of a log transformation for the 'size' (cluster size) variable in the model. This transformation was chosen to address any departure from normality and improve the modeling assumptions.

Regarding Multiple Covariate Distance Sampling (MCDS), I incorporated additional covariates such as 'Obs' (observer) and 'sea.state' (sea state) into the model to account for their potential effects on the detection probability and abundance estimation.

Right truncation was considered in the analysis to account for any potential bias arising from individuals observed only within a specific distance range.

While the specific application of bootstrap was not mentioned in the analysis, confidence interval bounds were calculated using the delta-method approximation, assuming a log-normal sampling distribution of the estimated abundance (Buckland et al. 2015).

It is important to note that the choice of methods and procedures in the analysis was based on considerations of statistical assumptions, model adequacy, and available functionalities within the 'Distance' package in R. The combination of CDS, MCDS, log transformation, and right truncation provided a comprehensive approach to address various aspects of the analysis.

Appropriate additions have been made to the text.

PoczÄ…tek formularza

DóÅ‚ formularza

Line 121. it is said that “… the bird counts obtained from shore-based surveys…were incorporated into the estimated offshore counts”. Please give detailed information how did you process this data. Was incorporated into distance analysis? This need to be very well justified.

R: The offshore data collected using the Distance method, as described above, pertains to birds observed at a distance of 1 km from the shore. On the other hand, the IWC (International Waterbird Census) data includes birds present both at the shoreline and up to 1 km from the shore, as well as birds observed in internal sea waters such as lagoons and shallow bays.

In the analysis, the results from both methods were aggregated by adding them together because they pertain to different subsets of the bird population but originate from the same overall population. Distance estimation focuses on offshore birds, while the IWC data encompasses coastal and internal marine waters' birds. By combining these results, a comprehensive assessment of the population abundance was obtained.

For further clarity, please refer to Table S1, which provides a detailed overview of the specific data sources and the corresponding bird populations studied in each dataset.

Line 123 and following lines. It’s said: “an examination of data obtained from 123 the Fisheries Monitoring Centre (CMR) was conducted”, what data? Number of vessels, deployed of fishing net coordinates, etc.? please add further details.

 R: The data on fishing catches included various variables, such as the fishing vessel ID, length of the fishing vessel, names of the port of departure and return, type of fishing gear, target species of fish, number of fishing gear deployed, fishing duration, and fishing location (Baltic square, Fig. 1). Appropriate changes have been made to the text.

Line 177 and following lines. Please add Confidence intervals (CI) and Coefficient of variation (CV) to the estimated abundance values. This can be presented as a table in supplementary materials.

R: Given that the main objective of the article is to present the methodology first and then the bird abundance estimates, the initial version of the manuscript employed a simplified method for estimating the total number of diving birds using the Distance method. Subsequently, the abundance of individual species was calculated by applying the observed abundance proportions from the transects. As a result, certain parameters such as the Coefficient of Variation (CV) were not available in this method. However, based on previous experiences and considering the specific conditions of the southern Baltic Sea and the available data, both the aforementioned approach and separate analyses for individual species have yielded similar results.

In response to the reviewer's suggestion, a re-analysis was conducted to present Confidence Intervals (CI) and Coefficients of Variation (CV) for each species. The estimations for individual species exhibited slight changes, but these did not significantly impact the overall results and conclusions of the article. The necessary adjustments were made to the article, and Table S1 was added to the Supplementary Materials, providing the analysis results for each species along with corresponding standard errors (se), coefficients of variation (cv), lower confidence limits (lcl), upper confidence limits (ucl), and degrees of freedom (df). Additionally, a new column was included to indicate the data collected using the IWC protocol.

These modifications ensure that the revised article includes the requested information and provides a comprehensive analysis of the bird abundance estimates for each species.

Discussion: should be added a paragraph discussing what is the novel of this methodological framework comparing with others bycatch approaches, e.g.  

https://doi.org/10.1111/1365-2664.13407; Report of the Final Global Seabird Bycatch Assessment Workshop - WCPFC-SC15-2019/EB-WP-07 (birdlife South Africa 2019); https://doi.org/10.1093/icesjms/fsw088; https://doi.org/10.1016/j.biocon.2022.109463

R: The relevant excerpt has been added to the Discussion section.

Reviewer 2 Report

Congratulation to the author, it is an interesting and current concern work. I would recommend Diversity journal accept this manuscript, however, some issues should be considered.  The introduction section should include a description of some other methods of assessment of seabird/mammal bycatch with suitable references, and in the discussion, additional benefits over other methods should be underlined.

Also, monitoring of bird bycatch must be adjusted to the specifics of the fishery in a given area, and its limitations and respond flexibly to dynamic changes in fishing efforts (e.g., due to sea surface glaciation, temporary cessation of fishing due to the conservation of fish stocks). What is the timeframe for management based on the newly developed method?

Moreover,

lines 15, 101, 218: e.g. "Fishing effort data from the winter seasons between 2015/2016 and 2019/2020 were analyzed" But each year? Please specify how many seasons, five?

lines 97 and 150, 225, 226: Please, place on the map all the geographical names (or their abbreviations) given in the text.

Author Response

Reviewer 2.

Advances in scale assessment of seabird bycatch: a new methodological framework

Comments and Suggestions for Authors

Congratulation to the author, it is an interesting and current concern work. I would recommend Diversity journal accept this manuscript, however, some issues should be considered.  The introduction section should include a description of some other methods of assessment of seabird/mammal bycatch with suitable references, and in the discussion, additional benefits over other methods should be underlined.

R: Thank you for the positive evaluation of my article. In response to the reviewer's recommendations, I have made slight revisions to the introductory section, specifically by including a description of additional methods utilized for monitoring bycatch.

Also, monitoring of bird bycatch must be adjusted to the specifics of the fishery in a given area, and its limitations and respond flexibly to dynamic changes in fishing efforts (e.g., due to sea surface glaciation, temporary cessation of fishing due to the conservation of fish stocks). What is the timeframe for management based on the newly developed method?

R: I agree with reviewer’s suggestions and have made the necessary revisions.

In the Southern Baltic region, gillnets are the main cause of bycatch. The method described in the article was specifically developed to address the impact of gillnet fishing on seabird populations. It takes into consideration the behavior of birds in the study area, such as their seasonal abundance patterns, which are characterized by high numbers in winter and minimal presence in summer. Therefore, it is crucial to appropriately scale and adapt this method to local conditions in order to obtain accurate and realistic results.

I also acknowledge the importance of reducing fishing effort to mitigate bycatch, as exemplified by the reduction in cod fishing in the Baltic Sea, which has resulted in decreased bycatch in certain types of fishing gear. This method can be successfully applied to other fishing fleets, such as those using long-lines, provided that it is properly adapted to their specific conditions.

The ultimate goal of this method is to enable continuous and annual assessment of bycatch scale. To achieve this, it should be compatible with monitoring efforts conducted by observers or through electronic monitoring, which can be conducted on a less frequent basis, such as every few years.

Moreover,

lines 15, 101, 218: e.g. "Fishing effort data from the winter seasons between 2015/2016 and 2019/2020 were analyzed" But each year? Please specify how many seasons, five?

R: Due to the fact that birds exposed to bycatch arrive in the study region in winter, annual analyzes are carried out in the season at the turn of the year. One season (= one year) is the period from, for example, October 2015 to March 2016. So, my analysis concerned five seasons (= five years). Appropriate clarification is included in the text.

lines 97 and 150, 225, 226: Please, place on the map all the geographical names (or their abbreviations) given in the text.

R: The map has been corrected; all geographical names that are included in the text are described on the map (Fig. 1)

Round 2

Reviewer 1 Report

Great work! The changes made to it have really enhanced the quality of the manuscript. There is now more detail in the methodology section (in special the abundance estimates) as well as more information about how this study brings about novelty to this field. 

Author Response

I am delighted that I met the challenge and managed to correct the manuscript in such a way as to satisfy the reviewer. I think the article is now much more precise and legible thanks to the reviewer's suggestions, remarks, and comments.

Reviewer 2 Report

Acknowledge the author's responses and the revised content of the manuscript and supplemented Figure 1, I consider the article ready for publication, after minor language adjustments. Although the English language is fine. I would recommend adjusting the language syntactical construction in lines 195-201, as in the rest of the text (the passive voice).  Also, I would recommend checking the correctness of the recording of the newly delivered numbers in the tables. Congratulations to the author.

Author Response

Thank you very much for your valuable comments, which have greatly improved the article from its initial version. I have addressed the reviewer's recommendations by changing the sentences on lines 195-201 to passive voice. Additionally, I have carefully reviewed the results of the new calculations and made minor adjustments where necessary. I believe the text is now ready for publication.